Electrical stimulation promoting the angiogenesis in diabetic rat perforator flap through attenuating oxidative stress-mediated inflammation and apoptosis

Chen Cong
Li Xiaolu
Hu Yong
Chen Yuan
Wang Hongrui
Li Xian sdulixian@163.com
http://orcid.org/0000-0002-3260-6299 Li Xiucun triumphlixc@163.com
Second Hospital of Shandong University , Jinan, Shandong , China
Leppik Liudmila
Electronic publication date: 2024 Jan 31
Publication date: 2024
Volume: 12
Electronic Location ID: e16856
Received 2023 Aug 29; Accepted 2024 Jan 8
Copyright: © 2024 Chen et al.
Copyright year: 2024
Copyright holder: Chen et al.
License: This is an open access article distributed under the terms of the Creative Commons Attribution License, which permits unrestricted use, distribution, reproduction and adaptation in any medium and for any purpose provided that it is properly attributed. For attribution, the original author(s), title, publication source (PeerJ) and either DOI or URL of the article must be cited.
License URL: https://creativecommons.org/licenses/by/4.0/

Keywords: Electrical stimulation, Angiogenesis, Oxidative stress, Inflammation, Diabetic ischemic skin flaps

Funding: Natural Science Foundation of Shandong Province, China ZR2020MH195; ZR2022MH168 This work was supported by the Natural Science Foundation of Shandong Province, China (No. ZR2020MH195; ZR2022MH168). The funders had no role in study design, data collection and analysis, decision to publish, or preparation of the manuscript.

==============================
Background

Skin flap transplantation is one of the effective methods to treat the diabetes-related foot ulceration, but the intrinsic damage to vessels in diabetes mellitus (DM) leads to the necrosis of skin flaps. Therefore, the discovery of a non-invasive and effective approach for promoting the survival of flaps is of the utmost importance. Electrical stimulation (ES) promotes angiogenesis and increases the proliferation, migration, and elongation of endothelial cells, thus being a potential effective method to improve flap survival.

Objective

The purpose of this study was to elucidate the mechanism used by ES to effectively restore the impaired function of endothelial cells caused by diabetes.

Methods

A total of 79 adult male Sprague-Dawley rats were used in this study. Gene and protein expression was assessed by PCR and western blotting, respectively. Immunohistochemistry and hematoxylin-eosin staining were performed to evaluate the morphology and density of the microvessels in the flap.

Results

The optimal duration for preconditioning the flap with ES was 7 days. The flap survival area percentage and microvessels density in the DMES group were markedly increased compared to the DM group. VEGF, MMP2, and MMP9 protein expression was significantly upregulated. ROS intensity was significantly decreased and GSH concentration was increased. The expression of IL-1β, MCP‑1, cleaved caspase-3, and Bax were downregulated in the DMES group, while TGF-β expression was upregulated.

Conclusions

ES improves the angiogenesis in diabetic ischemic skin flaps by attenuating oxidative stress–mediated inflammation and apoptosis, eventually increasing their viability.

Introduction

Diabetes mellitus (DM) is a common chronic metabolic disease characterized by hyperglycemia (Tomic, Shaw & Magliano, 2022). DM is linked to a substantial risk of foot complications, such as ulceration, gangrene, and consequent amputation. Amputation due to diabetes-related foot ulceration represents a significant majority (up to 85%) of non-traumatic amputations, with an annual incidence of ulceration of approximately 2% and a lifetime incidence of 34% (Armstrong, Boulton & Bus, 2017; Tomic, Shaw & Magliano, 2022). Currently, skin flap transplantation is considered one of the effective methods to treat ulcers (Lee et al., 2014). Kim et al. (2021) reported that the incidence of flap necrosis used to repair wounds in diabetic patients (50%) is significantly greater than in non-diabetic patients (13.3%). Additionally, Demiri et al. (2020) demonstrated that the necrotic rate of the ischemic skin flaps in diabetic patients reaches 35.2%. Despite preoperative measures such as vessel skeletonization and flap viability assessment, flap necrosis is still the prevailing and consequent complication of ischemic skin flaps under diabetes (Demiri et al., 2020; Kim et al., 2021).

Diabetes-related metabolic dysfunction leads to the overproduction of mitochondrial superoxide and other reactive oxygen species (ROS) in the endothelial cells (ECs) of both macro- and microvasculature (Giacco & Brownlee, 2010; Kohnert, Freyse & Salzsieder, 2012). The hyperglycemic state associated with diabetes leads to an increase in the levels of markers indicating DNA damage induced by oxidative stress, such as 8-hydroxy-2’-deoxyguanosine (8-OHdG) (Oguntibeju, 2019). Hyperglycemia-induced oxidative stress increases the expression of pro-inflammatory cytokines, with infiltrating macrophages releasing inflammatory cytokines that subsequently trigger local and systemic inflammation (Wellen & Hotamisligil, 2005). Increased intracellular ROS results in decreased angiogenesis under ischemia, various proinflammatory pathways are activated, and cell apoptosis increases, ultimately resulting in vascular damage (Giacco & Brownlee, 2010; Vecchié et al., 2019; Lee, Yun & Ko, 2022). Furthermore, increased oxidative stress and enhanced inflammatory responses are contributing factors of the development of diabetic vascular dysfunction, particularly in relation to microvascular complications (Shi & Vanhoutte, 2017). Additionally, DM negatively impacts endothelial cell function, resulting in a decrease in the expression of vascular endothelial growth factor (VEGF) in the skin (Costa et al., 2013; Shi & Vanhoutte, 2017). Diabetic individuals suffer from vasculopathy, characterized by both macrovascular and microvascular injuries, leading to reduced blood perfusion in the flaps and an increased necrotic area (Kim et al., 2021). Thus, the primary cause for the high rate of necrosis in ischemic flaps among diabetic patients may be attributed to the intrinsic damage to vessels in DM (Kim et al., 2021).

To date, surgical flap delay remains the most effective means of enhancing skin flap survival area, but it is an invasive two-step procedure associated with several complications (Gersch et al., 2017; Li et al., 2019). Other techniques are available to enhance the viability of the skin flap, such as venous super-drainage (Wang et al., 2020a; Zhu et al., 2023), and arterial supercharging (Fang et al., 2020; Wang et al., 2020a), but they come with disadvantages such as significant tissue trauma, prolonged surgical duration, and the requirement for advanced microvascular anastomosis skills. Pharmacologic preconditioning, also known as chemical delay, increases the survival area of the skin flap. However, it is important to note that chemical drugs are frequently toxic and may induce other undesirable reactions (Temiz et al., 2016; Chen et al., 2019b). Despite attempts to enhance angiogenesis through pharmacological means to improve flap survival, no effective drug for surgical flap delay has yet been discovered to regulate this complex process (Doğan & Özyazgan, 2015). Thus, the search of a non-invasive and effective method for managing ischemic necrosis in diabetic ischemic flap transplantation has emerged as a pressing concern in the field of flap surgery. Electrical stimulation (ES) is considered a promising approach for promoting tissue regeneration (da Silva et al., 2020), such as bone tissue engineering treatments (Mobini et al., 2017; Leppik et al., 2020), ES-induced vascularization in tissue (Wang & Meng, 2023), and wound healing (Rabbani et al., 2023). In addition, ES is extensively used in clinical settings as a rehabilitation method, effectively mitigating muscle atrophy and alleviating pain (Allen et al., 2023).

Previous study demonstrated that ES in rat ischemic limb contributes to a significant increase in the expression of vascular endothelial growth factor (VEGF) and an evident enhancement in capillary density (angiogenesis) in the stimulated muscle (Kanno et al., 1999). Jeong et al. (2017) observed that ES accelerates the formation of capillaries and arterioles in the ischemic area of athymic mice hindlimb, leading to a reduced muscle necrosis and fibrosis, ultimately inhibiting the necrosis of the ischemic hindlimb. It is worth mentioning that vascular ECs play a crucial role in the process of angiogenesis. Previous in vitro studies showed that ES (electric field: 150–400 mV/mm) enhances the production of VEGF and membrane metalloproteinases (Tzoneva et al., 2016). Additionally, ES induces favorable physical changes in ECs, including increased cell proliferation, elongation, altered cell shape, reorientation of the long axis of the cell, alignment, and directional cell migration (Chen et al., 2019a; Cunha, Rajnicek & McCaig, 2019; Geng et al., 2019; Luo et al., 2021). These alterations are essential prerequisites for the occurrence of angiogenesis. Furthermore, Long et al. (2019) performed a study using a rat model of middle cerebral artery occlusion ischemia-reperfusion injury, and they observed an evident decrease in the level of malondialdehyde (MDA) in the ES group. Conversely, the activity of glutathione (GSH) and superoxide dismutase (SOD) was significantly increased in comparison to that in the middle cerebral artery occlusion group (Long et al., 2019).

Although Doğan & Özyazgan (2015) demonstrated that the use of ES in the normal area prior to skin flap elevation increases the blood flow following the skin flap elevation and increase the skin flap viability compared to that in the control group, but their findings did not provide sufficient evidence to elucidate the underlying mechanism of increased vascularity. On this basis, our research focused on ischemic skin flaps of diabetes, and aimed to elucidate the mechanism used by ES to effectively restore the impaired function of endothelial cells caused by diabetes, as well as to explore strategies for enhancing angiogenesis in the perforator flap. We found that ES promoted angiogenesis in diabetic rat perforator flaps by attenuating oxidative stress-mediated inflammation and apoptosis, increasing the viability of the multi-territory perforator flap.

Materials and Methods

Animal and study protocols

This study was approved by the research ethics committee of the Second Hospital of Shandong University (KYLL-2019(KJ)A-0203). The experiments were performed according to the National Institutes of Health Guide for the Care and Use of Laboratory Animals. In addition, this research also followed the “Animal Research: Reporting In Vivo Experiments” (ARRIVE) Guidelines. A total of 79 healthy adult male Sprague–Dawley rats (weight: 300–350 g) were purchased from Jinan Pengyue Experimental Animal Co., Ltd (Jinan, China). Each cage contained one rat, rats were housed under standard conditions (humidity: 45–55%, temperature: 23–25 °C, and 12 h light/dark cycle) and fed with food and tap water at libitum. The feces were removed from the cages every 3 days. All rats were anesthetized before the experiment using 2% pentobarbital sodium (40 mg/kg, intraperitoneal injection) after isoflurane induction. All rats were euthanized by an intraperitoneal injection of pentobarbital sodium overdose (150 mg/kg) at the end of the experiment.

The rats with the direct anastomotic vessels in the flap donor site were excluded prior to the experiment. This study was performed in two parts. Part one consisted of the evaluation of the optimal duration of the ES preconditioning flap, performed using 28 normal rats randomly divided into the four groups: control group (no ES), ES 3-day group, ES 5-day group, and ES 7-day group. Part two consisted of the use of ES to improve the multi-territory flap viability by the induction of angiogenesis in diabetic rats, performed using 51 rats randomly divided into three groups: 17 normal rats used as control group (control: no ES), 17 diabetic rats as the DM group (DM, no ES), and 17 diabetic rats pretreated with ES used as the DM ES group (DMES). Flap viability, histological analysis and molecular biology analysis were performed in each group.

Animal model

Vascular anatomy: according to a previous report (Li et al., 2019), the flap can be classified into five zones: deep circumflex iliac (DCI) artery angiosome, proximal choke zone (PCZ), posterior intercostal (PIC) artery angiosome, distal choke zone (DCZ), and thoracodorsal (TD) artery angiosome from the caudal to the cranial part (Fig. 1A). Based on the anatomical theory between adjacent angiosomes (Cormack & Lamberty, 1984), the DCI is an anatomical territory, the PIC is a dynamic territory, and the TD is a potential territory in this flap. Thus, DCZ was selected for histological and molecular biology examination.

Figure 1 Flap model, and scheme of the electrostimulation equipment.

(A) Vascular anatomy: the perforated vessels of the three angiosomes were observed following the flap elevation, and three angiosomes and two choke zones were present in the dorsal flap, including the deep circumflex iliac (DCI), proximal choke zone (PCZ), posterior intercostal (PIC), distal choke zone (DCZ), and thoracodorsal (TD) vessels. (B) Flap design. (C) The flaps were raised based on the DCI artery perforator. (D) Scheme of the electrostimulation equipment: the equipment comprises two electrode stickers, each measuring 4.0 × 4.0 cm, strategically affixed to the PCZ and DCZ of the flap, respectively.

Flap design: after shaving, a rectangular flap of approximately 11 cm × 3 cm in dimension over the unilateral dorsum of the rat was designed based on the DCI artery perforator vessels (Fig. 1B). The caudal demarcation was situated at the superior edge of the musculus gluteus maximus and its cranial demarcation at the 7th cervical spinous process; the medial demarcation of the flap was located at the midline of the spine.

Flap harvest: all surgical procedures were performed under standard sterile conditions. The skin flap was undermined above the panniculus carnosus, and the perforators of the TD and PIC arteries were ligated and cut off. The flaps were raised based on the DCI artery perforator vessels (Fig. 1C) and then were sutured in situ using 4–0 nonabsorbable suture.

Generation of the DM rats

Sprague–Dawley rats were fed with tailor-made high-sugar and high-fat fodder for 4 weeks. Afterwards, streptozotocin (50 mg/kg) dissolved in citrate buffer (pH = 4.5) was intraperitoneally injected for 2 days. Blood samples were collected form the tail vein to measure the random plasma glucose. The random plasma glucose value was ≥16.7 and ≤33.3 mmol/L, suggesting that the diabetic rat model was successfully constructed (Tam et al., 2014). A total of 34 diabetes rats were generated.

ES

After completing the flap design, both electrodes stickers of the electrostimulation equipment (Xiangyu Medical Equipment Co. Ltd., Anyang, China) placed the PCZ and DCZ of the flap respectively (Fig. 1D). ES was performed under isoflurane anesthesia for 40 min daily (Fig. 1D; Frequency: 1,000 Hz, current intensity: 10 mA).

Flap viability

The assessment of flap survival area was performed on day 7 after the surgery. Images of the skin flaps were captured using a digital camera (Canon EOS800D; Canon, Tokyo, Japan), and Adobe Photoshop CS6 imaging analysis was used to determine the survival area. This measurement was expressed as a percentage by dividing the survival area by the total area of the flap and multiplying by 100%.

Histological analysis

The skin flap was raised at the scheduled time and flattened on the wooden board under a standard condition to maintain its accurate size. The flap was photographed using a digital camera (Canon EOS800D; Canon, Tokyo, Japan) to allow the vascular morphological analysis, using a plastic rule as a reference.

A tissue specimen measuring 3 × 0.5 cm was obtained from the DCZ for the purpose of histological analysis. The specimen was fixed in a solution of neutral buffered formalin for a duration of 24 h, after which it underwent dehydration. Subsequently, the tissue was embedded in paraffin and sectioned into slices measuring 3 μm in thickness. Hematoxylin and eosin (H&E) staining, as well as immunohistochemical (IHC) staining, were conducted. The stained sections were then captured using the Nanozoomer Digital Pathology (NDP) scanner S60 (NDP Scan C13210-01; Hamamatsu Photonics K.K., Hamamatsu, Japan) in order to convert the images into the NDP format.

The quantification of vessels with a diameter (D) greater than 0.1 mm (circumference [C] exceeding 0.314 mm; C = πD) in the subdermis and muscle layers was conducted through the utilization of H&E staining. The measurement of vascular circumference was accomplished using the NDP view software on the NDP image.

Microvessels density (N/mm2) in the subdermis was assessed through IHC staining of CD31 (rabbit anti-rat CD31, #: ab281583, 1:100 dilution; Abcam). Any aggregation of brown endothelial cells was regarded as a single quantifiable microvessel. Five randomly chosen high power fields (20× magnification, 0.85 × 0.48 mm/field) were examined, and the microvessel count was determined for each NDP image.

DNA damage by oxidative stress was assessed using the IHC staining of 8-OHdG antibody (#ab48508; Abcam, Cambridge, MA, USA; 1:200 dilution). Apoptosis was measured by the IHC staining of cleaved caspase-3 (CC3; #9664; CST, Danvers, MA, USA; 1:1,000 dilution). Any brown endothelial cell cluster was considered as a single countable positive cell. Five high power fields (20× magnification, 0.85 × 0.48 mm/field) including the vessels were randomly selected in the subdermis in each slice, and the number of positive cells and total cells were counted in each field.

Western blotting

A tissue sample of 3 × 1 cm in size was collected from the DCZ. The sample was lysed using radioimmunoprecipitation assay lysis buffer to extract the total proteins, which were subsequently separated by gel electrophoresis using gels at different concentrations (range, 8 to 15%) and transferred onto a polyvinylidene fluoride membrane.

The membrane was incubated with the following primary antibodies at 4 °C overnight: VEGF (#sc-7269, 1:500 dilution; Santa Cruz Biotechnology), matrix metalloproteinase-2 (MMP2: ab92536, 1:1,000 dilution; Abcam), matrix metalloproteinase-9 (MMP9: ab76003, 1:1,000 dilution; Abcam), monocyte chemoattractant protein-1 (MCP-1: #AF7437, 1:500 dilution, Beyotime), transforming growth factor β (TGFβ: #AF0297; 1:1,000 dilution; Beyotime), Bax (#14796, 1:500 dilution; Cell Signaling Technology, Danvers, MA, USA), β-actin (#AC026; 1:5,000 dilution; ABclonal, Wuhan, China). The membrane was treated with the corresponding secondary antibody and the bands were visualized using the automatic chemiluminescence/fluorescence image analysis system (Tanon 4800). The intensity of the bands was assessed by ImageJ2 software (NIH, Bethesda, MD, USA) and represented as fold change relative to that of β-actin.

Reverse transcription-polymerase chain reaction (RT-PCR)

Total RNA was extracted from the DCZ tissue. Reverse transcription was performed to obtain cDNA. qPCR was carried out using SYBR® Premix Ex Taq (TaKaRa) and the fluorescence qPCR instrument (ABI QuantStudio 5). Each amplification was performed in triplicate. Relative gene expression was calculated by the 2–ΔΔCT method, using β-actin as the normalization control. The primer sequences used in this study were the following: interleukin-1β (IL-1β): CCAGGATGAGGACCCAAGCA (forward), TCCCGACCATTGCTGTTTCC (reverse);β-actin:CACCATGTACCCAGGCATTG (forward), TCGTACTCCTGCTTGCTGAT (reverse).

Determination of ROS and GSH

A tissue sample of 3 × 0.5 cm in size was isolated from the DCZ and mechanically homogenized. The supernatant was collected to measure the amount of ROS and glutathione peroxidase (GSH) according to the manufacturer’s instructions. Tissue ROS intensity was evaluated using the tissue ROS assay kit (#BB-460512, Bestbio, Shanghai), and expressed as florescence intensity (RFU)/protein concentration (μg/L). Tissue GSH concentration (μmol/g protein) was measured using the reduced GSH assay kit (#A006-2-1; Nanjing Jiancheng Bioengineering).

Tissue GSH concentration (μmol/g protein) was calculated as follows = [(Measured OD value- Blank OD value)/(Standard OD value-Blank OD value)] × standard sample concentration (20 μmol/L) × dilution fold (two folds) ÷ protein concentration (g prot/L).

Statistical analysis

Statistical analysis was performed using GraphPad Prism 9. All measurements were performed by two researchers in a double-blind manner. Results were expressed as mean ± standard deviation. The gaussian distribution was evaluated using the Shapiro-Wilk test. Groups were compared using the t-test or Mann-Whitney U test. A value of P < 0.05 was considered statistically significant.

Results

Optimal duration of the ES preconditioning flap

The flap survival area percentage was 80.3% in the control group, 87.3% in the 3-day group, 88.2% in the 5-day group, 94.0% in the 7-day group (Fig. 2; original data: Table S1). The flap survival area percentage in the 7-day group was significantly increased in contrast to that in the control group (Fig. 2, P = 0.0026). Therefore, the optimal duration of ES preconditioning flap was 7 days, since it corresponded to the maximum survival area (Fig. 2).

Figure 2 Optimal duration of electrical stimulation (ES) preconditioning flap.

Control group: no ES; 3-day group: ES duration for 3 days; 5-day group: ES duration for 5 days; 7-day group: ES duration for 7 days. NS: not significant. (N = 7 per group).

ES increased the diabetes flap viability

The flap survival area percentage was 80% in the control group, 72% in the DM group, and 92% in the DMES group (Fig. 3A). The statistical analysis revealed that the flap survival area percentage in the DM group was significantly less than that in the control group (Fig. 3B, P = 0.0043; original data: Table S2), while the flap survival area percentage in the DMES group was remarkably more than that in the control group (Fig. 3B, P = 0.0022; original data: Table S2) and DM group (Fig. 3B, P = 0.0022; original data: Table S2).

Figure 3 Electrical stimulation (ES) significantly improves diabetes ischemic skin flap viability.

(A) Flap survival area; Black circle: flap necrotic area. (B) Flap survival area percentage. Control: no ES, DM: diabetes mellitus group (no ES), DMES: diabetes mellitus ES group.

The vascular morphological changes showed that the vascular network of the DM group was much sparser than that of the control group, while the vascular network of the DMES group was much denser than that of the control and DM group (Fig. 4A). The results of the H&E staining revealed that the number of vessels greater than 0.1 mm in diameter in the DM group was significantly lower than that in the control group (Fig. 4B, P = 0.024; original data: Table S3), while the number of vessels greater than 0.1 mm in diameter in the DMES group was markedly greater than that in the control group (Fig. 4B, P = 0.023; original data: Table S3) and DM group (Fig. 4B, P = 0.003; original data: Table S3). IHC staining of CD31 showed that CD31-positive microvessels density in the DM group was less than that in the control group, while CD31-positive microvessels density in the DMES group was more than that in the control and DM group (Fig. 5; original data: Table S4).

Figure 4 Electrical stimulation (ES) significantly increases the number of vessels with a vascular diameter > 0.1 mm.

(A) Changes in vascular network. (B) H&E staining results: number of vessels with a vascular diameter > 0.1 mm. Control: no ES; DM: diabetes mellitus group (no ES); DMES: diabetes mellitus ES group (N = 4 rats per group).

Figure 5 Electrical stimulation (ES) significantly increases the microvascular density.

(A–C) Immunohistochemical staining of CD31 in the control (A), diabetes mellitus (B) and diabetes ES (C) group. (D) CD31-positive microvessels density. Control: no ES; DM: diabetes mellitus group (no ES); DMES: diabetes mellitus ES group (N = 4 rats per group).

MMP-2 and MMP-9 are widely considered as the main proteolytic enzymes involved in the degradation of the extracellular matrix in the vascular basement membrane (Zhang et al., 2020). The proangiogenic factor VEGF activates MMP2 and MMP9, regulating the remodeling of the endothelial extracellular matrix, promoting the degradation of the EC basement membrane, favoring the migration of ECs and creating an advantageous environment to tubule formation (Zhang et al., 2020). It is worth noting that the expression of MMP-2 and MMP-9 is consistent with the change of neovascularization (Zhang et al., 2020). Thus, MMP2 and MMP9 were considered as angiogenic markers and measured. Western blotting showed that VEGF protein expression in the DM group was significantly reduced compared to the control group (1.0 ± 0.14 vs 0.42 ± 0.16, P = 0.0306; Fig. 6A), while VEGF protein expression in the DMES group was significantly increased compared to the DM group (1.3 ± 0.43 vs 0.42 ± 0.16, P = 0.0029; Fig. 6A; original band: Fig. S1). MMP2 protein expression in the DMES group was significantly upregulated compared to the control group (1.4 ± 0.3 vs 1.0 ± 0.08, P = 0.0240; Fig. 6B; original band: Fig. S1) and DM group (1.4 ± 0.3 vs 0.76 ± 0.1, P = 0.0018; Fig. 6B; original band: Fig. S1). MMP9 protein expression in the DMES group was much higher than in the control group (2.13 ± 0.21 vs 1.0 ± 0.45, P = 0.0014; Fig. 6C; original band: Fig. S1) and DM group (2.13 ± 0.21 vs 0.71 ± 0.17, P = 0.0003; Fig. 6C; original band: Fig. S1). These results indicated that DM significantly reduced vascular density and ES increased vascular density by the improvement of angiogenesis.

Figure 6 Western blotting showing the expression and quantification of the optical density of VEGF (A), MMP2 (B), and MMP9 (C) in the control, DM and DMES group.

Gel electrophoresis was performed under the same experimental conditions, and tailored blots are shown. VEGF: vascular endothelial growth factor; MMP2: matrix metalloproteinase-2; MMP9: matrix metalloproteinase-9; Control: no ES; DM: diabetes mellitus group (no ES); DMES: diabetes mellitus ES group (N = 4 rats per group). *P < 0.05, **P < 0.01, ***P < 0.001, NS: No signficance.

The IHC staining of 8-OHdG was performed to measure the oxidative stress damage in the flap tissue among the control, DM and DMES group since it is a marker for DNA oxidation during oxidative stress (Wang et al., 2020b). The results showed that the percentage of positive 8-OHdG in vascular endothelial cells of the flap tissue in the DM group was markedly increased compared to that in the control and DMES group, while no significant difference was found between the control and DMES group (Fig. 7A). In addition, the overproduction of ROS is the main cause of oxidative stress; thus tissue ROS intensity was also evaluated. ROS intensity in the DM group was much higher than that in the control group (1.8 ± 0.25 vs 0.38 ± 0.01, P < 0.0001; Fig. 7B; original data: Table S5) and DMES group (1.8 ± 0.25 vs 0.35 ± 0.04, P < 0.0001; Fig. 7B; original data: Table S5), while no significant difference was observed between the control and DMES group (0.38 ± 0.01 vs 0.35 ± 0.04, P = 0.9733; Fig. 7B; original data: Table S5). GSH is an endogenous antioxidant enzyme that protects the tissue from oxidative damage. GSH concentration in the DMES group was significant higher to that in the control group (3.5 ± 0.48 vs 2.7 ± 0.12, P = 0.0480; Fig. 7C; original data: Table S6) and DM group (3.5 ± 0.48 vs 1.5 ± 0.28, P = 0.0007; Fig. 7C; original data: Table S6), and GSH concentration in the DM group was much less than that in the control group (1.5 ± 0.28 vs 2.7 ± 0.12, P = 0.0114; Fig. 7C; original data: Table S6). The pro-inflammatory cytokines IL-1β and MCP‑1, and the anti-inflammatory cytokine TGFβ were measured. RT-PCR results showed that IL-1β gene expression in the DM group was much higher than that in the control group (2.0 ± 0.24 vs 1.0 ± 0.37, P = 0.0427; Fig. 7D; original data: Table S7) and DMES group (2.0 ± 0.24 vs 0.90 ± 0.51, P = 0.0341; Fig. 7D; original data: Table S7), while no significant difference was found between the control and DMES group (1.0 ± 0.37 vs 0.90 ± 0.51, P = 0.8871; Fig. 7D; original data: Table S7). Western blotting showed that MCP-1 protein expression in the DM group was much higher than that in the control group (2.38 ± 0.53 vs 1.0 ± 0.31, P = 0.0021) and DMES group (2.38 ± 0.53 vs 1.40±0.31, P = 0.0163), while no significant difference in MCP-1 protein expression was observed between the control and DMES group (Figs. 7E and 7G; original band: Fig. S2). TGFβ protein expression in the DM group was significantly downregulated compared to the control group (0.43 ± 0.16 vs 1.0 ± 0.36, P = 0.0282) and DMES group (0.43 ± 0.16 vs 1.0 ± 0.19, P = 0.0276), while no significant difference in TGFβ protein expression was found between the control and DMES group (Figs. 7F and 7H; original band: Fig. S2). These results indicated that ES attenuated oxidative stress-mediated inflammation.

Figure 7 Electrical stimulation (ES) attenuates oxidative stress-mediated inflammation.

(A) Immunohistochemical staining of 8-hydroxy-2’-deoxyguanosine (8-OHdG); black arrow: positive cells. (B) Reactive oxygen species (ROS) intensity in the tissue (N = 3 per group). (C) Glutathione (GSH) concentration in the tissue (N = 3 per group). (D) Interleukin-1β (IL-1β) gene expression (N = 3 per group). (E and H) Expression and quantification of the optical density of MCP-1 (E and G), and TGFβ (F and H) in the control, DM and DMES group (N = 4 per group) by western blotting. Gel electrophoresis was performed under the same experimental conditions, and tailored blots were shown. *P < 0.05, **P < 0.01, ***P < 0.001, ****P < 0.0001. NS: No signficance.

The expression of the pro-apoptotic proteins cleaved caspase-3 and Bax were measured. Caspase-3 is a pivotal regulatory protein of apoptosis and nuclear changes in apoptosis. It is cleaved to form cleaved caspase-3, which regulates the morphological and biochemical alterations in apoptosis (Silva et al., 2022). The immunohistochemical staining of cleaved caspase-3 showed that the percentage of positive cleaved caspase-3 in vascular endothelial cells of the flap tissue in the DM group was significantly increased compared to that in the control and DMES group, while no significant difference was observed between the control and DMES group (Fig. 8A). Western blotting showed that Bax protein expression in the DM group was significantly upregulated compared to the control group (1.6 ± 0.32 vs 1.0 ± 0.19, P = 0.0181) and DMES group (1.6 ± 0.32 vs 0.64 ± 0.42, P = 0.0142), while no significant difference in Bax protein expression was found between the control and DMES group (Figs. 8B and 8C; original band: Fig. S3).

Figure 8 Electrical stimulation (ES) attenuates apoptosis.

(A) Immunohistochemical staining of cleaved caspase-3. (B and C) Expression and quantification of the optical density of Bax in the control, DM and DMES group (N = 4 per group) by western blotting. Gel electrophoresis was performed under the same experimental conditions, and tailored blots were shown. *P < 0.05, **P < 0.01, NS: No signficance.

Discussion

Our findings demonstrated that ES significantly upregulated the expression of VEGF, MMP2, and MMP9 in the ischemic skin flaps of diabetic patients, increased the microvessels density and the number of vessels with a diameter longer than 0.1 mm, eventually increasing the survival area percentage of the ischemic skin flaps. Further investigation revealed that ES significantly upregulated the expression of the antioxidant GSH and the anti-inflammatory cytokine TGFβ in the ischemic skin flaps of diabetic patients. Moreover, ES markedly downregulated the expression of the oxidative injury markers ROS, pro-inflammatory cytokines IL-1β and MCP‑1, as well as pro-apoptotic proteins cleaved caspase-3 and Bax. Although Doğan & Özyazgan (2015) demonstrated that the use of ES in the normal area prior to skin flap elevation increases the blood flow following the skin flap elevation and increase the skin flap viability compared to that in the control group, our research focused on ischemic skin flaps of diabetes and reveal the mechanism of ES promoting angiogenesis in this specific pathological microenvironment.

ES is a non-invasive and non-pharmacological physical stimulus. At the molecular level, it is able to enhance the transportation of biomolecules, whether charged or uncharged, across biological membranes through the electrophoresis and electroosmosis (Gratieri, Santer & Kalia, 2017). At the cellular level, ES has an effect on a diverse range of cellular components, including ion channels, membrane-bound proteins, cytoskeleton, and intracellular organelles (Zhao, Mehta & Zhao, 2020). The underlying mechanisms responsible for the reaction of the cell to ES are currently being extensively investigated. Several hypotheses have been proposed, including the disruption of structural water, electroosmotic fluid flow, asymmetric ion flow and the activation of voltage-gated channels, mechanosensation, as well as redistribution of membrane components and lipid rafts (Zhao, Mehta & Zhao, 2020). These interactions ultimately lead to alterations in cellular behavior and functions, such as migration, contraction, orientation, and proliferation (Zhao, Mehta & Zhao, 2020).

The most remarkable morphological changes in the microvasculature of diabetic individuals are a thickened capillary basement membrane, reduced luminal dimension and number of capillaries, as well as the degeneration of pericytes (Sharma, Schaper & Rayman, 2020). The loss of autoregulation in microvascular blood flow leads to an increase in capillary pressure, which may trigger inflammatory responses within the microvascular endothelium. This results in endothelial injury and subsequent thickening of the capillary basement membrane, arteriolar hyalinosis, and reduced vasodilatory capacity (Sharma, Schaper & Rayman, 2020). In our experiments, the vascular density of the flap of the DM group was significantly reduced compared to that of the control group, and the expression of the angiogenetic protein VEGF was also remarkably downregulated. The positive 8-OHdG rate in vascular endothelial cells and ROS intensity in the DM group was increased. The concentration of GSH, which works as a protective agent against oxidative damage to the tissue, was decreased. The expression of the pro-inflammatory cytokines IL-1β and MCP‑1 was upregulated and the expression of the anti-inflammatory cytokine TGFβ was downregulated. The expression of the pro-apoptotic proteins cleaved caspase-3 and Bax was upregulated. The electron transport chain in the mitochondria is impaired in a chronic high glucose environment, resulting in the generation of ROS, stimulating the increase of proton leakage and altering the mitochondrial membrane potential, subsequently causing the release of cytochrome c, ultimately leading to apoptosis (Zhang et al., 2023). In short, oxidative stress-mediated inflammation induced by hyperglycemia was increased, thereby exacerbating cellular apoptosis, and reducing angiogenesis, ultimately leading to a decrease in the viability of the multi-territory perforator flap.

However, our results showed that the optimal duration for preconditioning the flap with ES was 7 days. ES significantly enhanced the survival area percentage of the diabetic ischemic skin flaps. Vascular density (microvessels density and number of vessels with a diameter longer than 0.1 mm) significantly increased in the DMES group compared to that in the DM group, and the angiogenesis-related protein expression (VEGF, MMP2, MMP9) was significantly upregulated. Li et al. (2023) performed a study in which they created a set of cell-scale to perform experiments to examine the electro-mechanical coupling phenomenon in human umbilical vein endothelial cells (HUVECs) during angiogenesis. They found that the stimulation by an external electrical field (Eex) polarizes intracellular calcium ions, generating a rear-to-front concentration gradient in HUVECs, and establishing an internal electric field (Ein) opposed to the Eex. Cells affected by changes in local calcium ion actively contract the cytoskeleton to activate Piezo1 channels, leading to the influx of extracellular calcium ions and gradually establishing a balance between Ein and Eex. In addition, they found that the electro-mechanical coupling feedback loop guides pre-angiogenic activities, such as elongation and migration of HUVECs. The external electrical field also promotes the expression of the angiogenesis-related genes VEGF and eNOS in HUVECs.

Our experiments showed that the 8-OHdG positive cell rate and the ROS intensity in the DMES group were significantly decreased compared to those in the DM group, and GSH concentration was increased. Previous studies demonstrated that external suitable ES leads to the change in cell membrane potential (Zhao et al., 2004), resulting in the polarization of the cell, inducing influx of extracellular Ca2+ (Emerson & Segal, 2001; Li et al., 2023), triggering calcium signal cascade reaction, and reducing the level of the oxidative stress. ES also promotes the entrance of the NFE2-related factor 2 (Nrf2) into the nucleus, thus upregulating the production of the protective factors heme oxygenase 1 (HO-1) and NADPH quinone oxidoreductase 1 (NQO1) through the anti-oxidative stress signaling pathway Kelch-like ECH-associated protein 1 (Keap1)/Nrf2 (Weng et al., 2023). Sha et al. (2015) concluded that ES significantly enhances the enzyme activity of superoxide dismutase, GSH, and other endogenous oxidation free radical scavenging systems, inhibits lipid peroxidation in tissue cell membranes, reduces the malondialdehyde content, and plays a protective role in tissues. In addition, our results showed that IL-1β and MCP‑1 expression was downregulated, while TGF-β expression was upregulated. ES changes the function of inflammatory cells, leading to a reduction in the activity of extracellular matrix modifier enzyme and matrix metalloproteinase-1, inhibiting the secretion of inflammatory factors, while maintaining unchanged the immune cell count (Kim et al., 2019; Shin et al., 2019). Thus, ES attenuated oxidative stress-mediated inflammation in ischemic skin flaps of diabetic rats.

Interestingly, our results also revealed that the expression of the pro-apoptotic proteins cleaved caspase-3 and Bax in the DMES group was downregulated compared to the DM group, but the exact mechanism is unclear. ES mitigates apoptosis by regulating intracellular protein expression and activating several signaling pathways, such as the MAP kinase pathway, and the ERK signaling pathway (Zhao et al., 2021). Yang et al. (2014) demonstrated that ES mitigates the apoptosis of ischemic cardiomyocytes in rats by upregulating Bcl-2 gene expression and downregulating Bax gene expression, although the precise mechanism remains unclear. ES also promotes cell survival by inducing the opening of voltage gated calcium channels, which causes Ca2+ influx into the cell, resulting in the activation of AKT by Ca2+. Akt induces cell survival by activating the transcription factor NF-KB, which translocates to the nucleus to induce the transcription of pro-survival genes. NF-KB also activates the anti-apoptotic protein Bcl-2 and inhibits the tumor suppressor p53 (Love et al., 2018). In addition, ES is nothing more than a flow of electrons. This flow of electrons mimics the flow of ions in the biological system, mainly altering the membrane potential (Vmem) of the cells. This alteration in Vmem is the root cause of the inhibition of apoptosis. This is the key point explaining why the survival area was increased after application of ES.

In addition, the use of medium frequency current offers the advantage of diminishing skin impedance, thereby minimizing the peripheral dissipation of electrical energy and facilitating a deeper penetration into the muscle (De Oliveira et al., 2018). Also, medium frequency current elicits multiple nerve fiber action potentials per burst, resulting in firing rates that are multiples of the burst frequency (De Oliveira et al., 2018). Conversely, the application of high frequency current may induce muscle fatigue (Szecsi & Fornusek, 2014).

Conclusions

ES improves the angiogenesis in ischemic skin flaps of diabetic rats through the reduction of oxidative stress–mediated inflammation and apoptosis, eventually increasing the ischemic skin flap viability. Thus, ES is a non-invasive, inexpensive, and effective method to increase the survival area of ischemic skin flaps in diabetic individuals.

Supplemental Information

Supplemental Information 1 The original Data.

Click here for additional data file.

Supplemental Information 2 Original western blotting band figures.

Click here for additional data file.

Supplemental Information 3 Author Checklist - Full.

Click here for additional data file.

Additional Information and Declarations

Competing Interests

Author Contributions

Ethics

Data Availability

The authors declare that they have no competing interests.

Cong Chen performed the experiments, analyzed the data, prepared figures and/or tables, and approved the final draft.

Xiaolu Li performed the experiments, prepared figures and/or tables, and approved the final draft.

Yong Hu performed the experiments, prepared figures and/or tables, and approved the final draft.

Yuan Chen performed the experiments, prepared figures and/or tables, and approved the final draft.

Hongrui Wang performed the experiments, prepared figures and/or tables, and approved the final draft.

Xian Li conceived and designed the experiments, performed the experiments, analyzed the data, prepared figures and/or tables, and approved the final draft.

Xiucun Li conceived and designed the experiments, performed the experiments, analyzed the data, prepared figures and/or tables, authored or reviewed drafts of the article, and approved the final draft.

The following information was supplied relating to ethical approvals (i.e., approving body and any reference numbers):

The research ethics committee of the Second Hospital of Shandong University approved this study (KYLL-2019(KJ)A-0203).

The following information was supplied regarding data availability:

The raw data is available in the Supplemental Files.

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
