# Peer review of "Electrical stimulation promoting the angiogenesis in diabetic rat perforator flap through attenuating oxidative stress-mediated inflammation and apoptosis"

_PeerJ, doi:10.7717/peerj.16856_

## Round 0.1 · original submission · Major Revisions

Please, provide necessary corrections as suggested by reviewers and emphasise the use of electrical stimulation, and its role in attenuation of oxidative stress in the introduction and discussion.

**Language Note:** The review process has identified that the English language must be improved. PeerJ can provide language editing services - please contact us at copyediting@peerj.com for pricing (be sure to provide your manuscript number and title). Alternatively, you should make your own arrangements to improve the language quality and provide details in your response letter. – PeerJ Staff

Reviewer 1 ·

Basic reporting

Experimental design is clear. Professional article structure, results, figures, tables are in-line. Raw data shared.

Literature references are insufficient.

Novelty of research is unclear.

Writing needs minor improvements.

Experimental design

Well defined, relavant and meaningful.

Validity of the findings

Though results are statistically sound, the novelty of the research is still unclear.
Fatih Doğan & İrfan Özyazgan 2015 has also shown effect of ES on the flap survival area. Please explain how your study is different in the context of novelty.

Additional comments

General:
Introduction needs more details
Author cites few publications especially in the introductions. Multiples citations are required to support any claim
Author needs to do more reading on the effects of ES at cellular and molecular level.
Fatih Doğan & İrfan Özyazgan 2015 has also shown effect of ES on the flap survival area. Please explain how your study is different in the context of novelty.
Line 54: Citations missing
Line 54: What is full form of DM?
Line 56: Please cite more references.
Line 56-57: Citation missing
Line 64: DIABETES causes…
Line 67-68-69: Apart from Giacco and Brownlee 2010, are there any other publications, which claims overproduction of ROS during diabetes? Please cite them. Please check: Kohnert KD, Freyse EJ, Salzsieder E. Glycaemic variability, and pancreatic β-cell dysfunction. Curr. Diabetes Rev. 2012;8:345–354. doi: 10.2174/157339912802083513.
Line 82-83: Please describe current approach/treatments for addressing ischemic necrosis during ischemic flap transplantation with reliable references. Please describe why these available treatments are not 100% successful. Use this as a motivation to go for ES.
Line 84: Please cite reliable references supporting this view.
Line 84: Also shortly explain how ES promotes tissue regeneration and then make a link to your hypothesis.
Line 85: “promote” without s
Line 86: What are the effects of ES on membrane potential (Vmem) of vascular cells during angiogenesis. How it changes? Please describe briefly.
General: Materials and methods needs more details
Line 104: please replace the word 'optimal timing' with 'optimal duration'

Line 105-106-108: Replace twenty-eight as 28, seventeen as 17, and fifty-one as 51.
Line 120: ‘’11cm × 3 cm’’ Either remove the space before ‘cm’ or keep it. Keep it consistent throughout the paper
Line 130: Please write down full name of SD rats
Line 133-134: Please cite the reference from which this method of creating DM rats were inspired..
Line 139: Please explain why 40min, 1000 Hz and 10 mA was chosen? Please write 'Hz' not HZ
Line 142-143-144: Please simplify the whole sentence
Line 143: Equation should be written separately not within paragraph
Line 147: describe digital camera model, company etc...
Line 152: It should be H&E staining
Line 172: Write down full name of RIPA buffer
Line 184: Describe as Describe as ImageJ2 software (NIH, USA)
Line 197: Shanghai with S capital
Line 200-201-202: Equation should be written separately not within paragraph.
Line 211: Remove the word ‘’flap’’ after preconditioning
Line 214: Optimal ‘’duration’’
Line 220: It should be survival area ‘’percentage’’ not ‘’rate’’
Line 225: It should be ‘’H&E’’ staining
Line 226: it is not clear what author means by term ''number of diameter'’. Diameter itself mentions a number.
Line 293: Grammar improvement as: luminal dimension is diminished, number of capillaries are reduced, and pericytes are degenerated.
Line 301-303: Formation of this long sentence is too difficult to understand, Please write in understandable manner
Line 311: Replace word ‘’timing’’ with ‘’Duration’’
Line 331-332: Explain briefly, How does ES change the function of inflammatory cells at the molecular level?
In the discussion, authors talk about how ES improves survival area however fundamental mechanism of ES is still missing..ES is nothing but flow of electrons. This flow of electrons mimics flow of ions in the biological system and mainly alters membrane potential (Vmem) of the cells. This alteration in Vmem is the root cause to the inhibition in apoptosis. This is a key point why survival area is increased after application of ES

Authors need to discuss more on ES mechanism at cellular and molecular mechanism.
Figures and Legends:
Figure 2 legend: Optimal ‘’duration’’ instead of time
Figure 3 legend: Electrical stimulation significantly improves diabetes ischemic skin flap viability. (B) Survival area percentage
Figure 4 B legend: H&E staining
Figure 5. legend Electrical stimulation significantly increases the microvascular density.
Please improve scales in figure 4,5,7 and 8. Alternately, you can mention them in legend

Annotated reviews are not available for download in order to protect the identity of reviewers who chose to remain anonymous.

·

Basic reporting

The use of professional English in this article was acceptable. figures, tables and shared data are presented in good quality.

Unfortunately, there is a lack of background discussion on Electrical Stimulation for skin regeneration/ remodeling/graft integration, etc.There are numerous references that could be used for the rationalization of the use of ES in tissue regeneration, the choice of ES parameters, etc. Some of these references include publications from Leppik et al., Levin et al., Mobini et al., etc. For instance, in the introduction part, there are only three sentences about ES (lines 82-88) with no reference. I highly recommend that the authors explain the motivation for using ES (explain why ES might combat oxidative stress- which what hypothetical mechanism) and elaborate on previous attempts to heal diabetic ulcers with ES and/or skin flap integration and survival.

I also found a lack of explanation in the results and discussion sections. It was not easy to understand why MMP2 and 9 were measured as angiogenic markers. Again, there is a lack of background, context, and referencing.

Experimental design

This work is in the scope of the PeerJ. This work provides ethical certificates for use of animals.

The research question is stated clearly, but the rationale of using ES for controlling oxidative stress is not elaborated. The authors aim to test if ES can improve the survival of skin flab by decreasing the oxidative stress. however, they did not explain why ES with those particular parameters are used.

There is no figure, description or schematic of electrical stimulation device. I highly recommend that the authors add this information in the main article and incorporate it in figure 1.

To optimize the timing of stimulation, the authors described an experiment and reported the results in Figure 2. I wonder if they have conducted this experiment considering the control group (no electrical stimulation). If yes, that needs to be reported. If not, it needs to be conducted.

Validity of the findings

In the western blot data, there are variations in beta actin, that is used as housekeeping protein for normalization. beta-actin is not a good choice when one using electrical stimulation. Besides, the experiment was not run cleanly. Nevertheless, if authors can repeat the WB, I encourage them to do so.

Conclusions is too short. not reflecting on any particular aspect of this work

Additional comments

I In this study, the authors explored the impact of Electrical Stimulation (ES) on oxidative stress related to skin flap survival in diabetic cases. While the research question is clear and ethical certificates are provided, critical aspects are lacking. The rationale for using ES to control oxidative stress is not adequately explained, and specific parameters for ES are not justified. Moreover, the absence of a detailed description of the ES device is a significant gap. The timing optimization experiment lacks clarity regarding the absence of a control group, raising concerns about experimental design. Despite acceptable professional English, the absence of background discussion, lack of contextualization, and insufficient referencing weaken the study. The choice of MMP2 and 9 as angiogenic markers lacks justification, and variations in beta-actin normalization cast doubts on data reliability. The conclusions are disappointingly brief, lacking substantive reflections on the study's limitations or potential improvements. A comprehensive revision addressing these issues is essential to enhance the study's credibility and scientific impact."

---

## Round 0.2 · accepted · Accept

The authors have significantly improved the quality of the manuscript so that it could be accepted for publication. Congratulations!

Reviewer 1 ·

Basic reporting

Fine

Experimental design

Fine

Validity of the findings

Fine

Additional comments

comments addressed

·

Basic reporting

The authors have addressed almost all the comments and correctly applied changes. However, my concern regarding beta actin has been remained. Beta actin is not a proper house keeping gene, because ES changes the expression of actin.

Experimental design

ok

Validity of the findings

ok